# Clinico-Virological Outcomes and Mutational Profile of SARS-CoV-2 in Adults Treated with Ribavirin Aerosol for COVID-19 Pneumonia

**DOI:** 10.3390/microorganisms12061146

**Published:** 2024-06-05

**Authors:** Giulia Morsica, Emanuela Messina, Sabrina Bagaglio, Laura Galli, Riccardo Lolatto, Michela Sampaolo, Maxime Barakat, Robert J. Israel, Antonella Castagna, Nicola Clementi

**Affiliations:** 1Unit of Infectious Diseases, IRCCS San Raffaele Hospital, 20132 Milan, Italy; messina.emanuela@hsr.it (E.M.); bagaglio.sabrina@hsr.it (S.B.); galli.laura@hsr.it (L.G.); lolatto.riccardo@hsr.it (R.L.); castagna.antonella1@hsr.it (A.C.); 2Laboratory of Microbiology and Virology, IRCCS San Raffaele Hospital, 20132 Milan, Italy; michela.sampaolo@hsr.it (M.S.); clementi.nicola@hsr.it (N.C.); 3Bausch Health, Laval, QC H7A 0B5, Canada; maxime.barakat@bauschhealth.com; 4Bausch Health, Bridgewater, NJ 08807, USA; robert.israel@bauschhealth.com; 5Faculty of Medicine and Surgery, Vita-Salute University, 20132 Milan, Italy; 6Laboratory of Microbiology and Virology, Vita-Salute San Raffaele University, 20132 Milan, Italy

**Keywords:** COVID-19, pneumonia, ribavirin aerosol, treatment, outcome, viral load, whole-genome sequencing

## Abstract

The emergence of new SARS-CoV-2 variants can affect vaccine efficacy, laboratory diagnosis and the therapies already available, triggering interest in the search for antiviral agents for SARS-CoV-2 infections. Ribavirin (RBV) is a broad-spectrum antiviral with demonstrated in vitro activity against multiple viruses, including SARS-CoV-2. This retrospective study evaluated the dynamics and viral clearance of SARS-CoV-2 in hospitalised adult participants (PTs) with COVID-19 pneumonia who received an RBV aerosol within a compassionate use study. The impact of RBV on the clinical outcome and the mutational profile of SARS-CoV-2 was also assessed. The median RNA values measured in nine PTs included in this study decreased from baseline to discharge (at BL, threshold cycle (Ct) = 22.4, IQR 19.84–5.07; at discharge, Ct = 27.92, IQR 26.43–36.11), with a significant decline in the Ct value evaluated by Friedman rank ANOVA analysis, *p* = 0.032. Seven out of nine PTs experienced a clinical improvement, while two PTs deceased during hospitalisation. In PTs with a favourable outcome, the virus clearance rate at discharge was 28.6%. The cumulative clearance rate was 71.4% within 14 days from discharge. A mutational pattern after RBV was detected in three out of five PTs in whom whole-genome sequencing was available. Our findings suggest that RBV limits SARS-CoV-2 replication, possibly resulting in a favourable clinical outcome. Ribavirin may also contribute to the mutational spectrum of SARS-CoV-2.

## 1. Introduction

The virus responsible for the COVID-19 pandemic, SARS-CoV-2, is a beta coronavirus like severe acute respiratory syndrome coronavirus (SARS-CoV) and the Middle East respiratory syndrome coronavirus (MERS-CoV) [1]. Several patients with SARS-CoV-2 infection develop acute respiratory distress syndrome likely due to an excessive pro-inflammatory host response, including aberrant induction of inflammatory cytokines, and infection is associated with severe lung pathology with a fatal outcome in a non-negligible number of cases [2,3]. 

The global clinical experience with ribavirin for the treatment of COVID-19 pneumonia started with SARS-CoV, for which RBV was initially indicated based on the pathological similarity of SARS-CoV to acute respiratory syndrome, which requires a typical administration of RBV and corticosteroids [4,5]. 

Ribavirin as an inhalation solution is FDA-approved for the treatment of infants and young children with severe lower respiratory tract infections due to respiratory syncytial virus [6] and has been shown to be effective against different influenza viruses [7,8]. This method of drug administration usually results in a higher concentration of the drug at the site of infection than systematic administration does and may minimise systemic toxicities. 

Since the course of COVID-19 begins with the binding of inhaled SARS-CoV-2 to cells in the nasal epithelium and may progress to involvement of the upper respiratory tract and migration into the lower respiratory tract, inhaled treatments, such as ribavirin aerosols, may offer a therapeutic advantage because the compound is delivered to the site of infection [9]. Recently, the administration of RBV through an aerosol has been made available in Italy for patients with COVID-19 as part of a compassionate use program, and preliminary data [10] showed a potential benefit of administration of ribavirin aerosol in SARS-CoV-2 viral clearance, with no adverse effects reported during treatment.

From a molecular point of view, RBV can limit viral replication based on multimodal antiviral proprieties: first, RBV is a guanosine analogue, and the most straightforward mechanism of action is the inhibition of viral RNA synthesis. A second mechanism of action involves the competitive inhibition of the inosine monophosphate dehydrogenase (IMPDH) enzyme through specific binding to the IMPDH substrate (inosine-5-monophosphate), ultimately leading to decreased synthesis and lower levels of guanosine triphosphate (GTP). A third mechanism of action is inferred from data demonstrating that RBV triphosphate can be incorporated into the viral genome, inducing RNA viruses to error catastrophe and lethal mutagenesis. Ribavirin can interact with the viral genome by being incorporated as a substrate into the RNA molecule and by causing transition mutations (A-to-G and C-to-U) [11,12]. To investigate this possibility, Crotty and colleagues sequenced the capsid-coding regions of poliovirus grown in the presence of varying concentrations of ribavirin [13]. Sequencing revealed an increase in mutations, particularly the predicted transition mutations. RBV pairing with both cytosine and uracil, once incorporated into the genome, would drive an increasing number of mutations [14,15]. 

As with other RNA viruses, SARS-CoV-2 may present high intra-host adaptability. 

It is well known that during the COVID-19 pandemic, a higher incidence of SARS-CoV-2 infection in the poorest countries resulted in the emergence of more transmissible viral variants, often showing improved immune escape compared to previously circulating variants. Therefore, virus adaptability represents a key molecular aspect that drives the evolution of variants of concern with higher or lower pathogenicity. Additionally, the presence of mutations can affect vaccine efficacy, laboratory diagnosis, and therapies already available [16]. Antivirals target viral enzymes involved in viral intracellular replication or hamper the viral entry process [17]. However, due to the adaptive selection of certain mutations, several drugs can become less efficient or, in the worst scenarios, ineffective [18]. 

As detailed above, RBV may limit SARS-CoV-2 replication by multimodal mechanisms, possibly limiting the emergence of novel virus variants. 

Here, we describe the possible effect of RBV administered via aerosols on viral load and clinical outcomes in hospitalised participants with COVID-19 pneumonia.

## 2. Materials and Methods

### 2.1. Study Design and Participants

This retrospective study included data collected between July 2020 and August 2022 within a compassionate use study allowing treatment with RBV as an inhalation solution, USP, in adult participants (PTs) with a laboratory-confirmed SARS-CoV-2 infection, diagnosed with primary mild–moderate COVID-19 pneumonia assessed by clinical or radiographic evidence of lower respiratory tract disease and oxygen saturation ≥94% in room air (with other infectious/bacterial or fungal concomitant aetiologies excluded) (https://www.aifa.gov.it/programmi-di-uso-compassionevole-covid-19, accessed 29 March 2024) which also involved a semi-quantitative analysis of SARS-CoV-2 load in nasopharyngeal samples at the start of RBV treatment (baseline evaluation) and at the end of treatment (EOT), corresponding to a 6-day course of RBV treatment or discontinuation, and at discharge. 

### 2.2. Procedures

Aerosolisation of ribavirin was carried out using a high-efficiency jet nebuliser (Pari boy SX [PARI Respiratory Equipment, Inc.] flowing at a rate of ~8 to 10 L of air per minute. Ribavirin was administered via an aerosol (10 mL of 100 mg/mL for 30 min twice daily for at most 6 days for a 2000 mg total daily dose). Computed tomography scans were performed to assess the presence of pneumonia and evaluate the extent of parenchyma inflammation. From a descriptive point of view, our radiologists defined the degree of pulmonary involvement using a visual assessment expressed on a scale (<10%, 10–25%, 25–50%, 50–75%, >75%) [19]. 

### 2.3. Clinical and Laboratory Monitoring

Clinical findings including medical history and physical, laboratory, and radiological investigation results were entered into a predesigned database. 

The clinical status severity (CSS) was rated on a 7-point scale ranging from death (category 1) to discharge (category 7), as previously described [20]. Briefly, 1 = death; 2 = hospitalised, requiring invasive mechanical ventilation or extracorporeal membrane oxygenation; 3 = hospitalised, requiring non-invasive ventilation or use of high-flow oxygen devices; 4 = hospitalised, requiring low-flow supplemental oxygen; 5 = hospitalised, not requiring supplemental oxygen but requiring ongoing medical care (related or not to COVID-19); 6 = hospitalised, not requiring supplemental oxygen or ongoing medical care; and 7 = discharged). The change in clinical/respiratory parameters was evaluated according to the compassionate use study protocol: PaO_2_/FiO_2_ ≥ 300, peripheral capillary oxygen saturation (SpO_2_). These parameters were measured at baseline, at the end of RBV administration (EOT, day 6 of treatment), and at discharge. High-resolution CT scans were performed at baseline and discharge. Blood laboratory tests considered at baseline and discharge included white blood cell (WBC) count, total lymphocytes, ferritin (normal value, males 30–400 ng/mL, females 15–150 ng/mL), interleukin-6 (IL-6, normal value < 7 pg/mL), lactate dehydrogenase (LDH, normal value 125–220 U/L), C-reactive protein (normal value < 6 mg/dL), fibrinogen, (normal value 150–400 ng/dL), and D-dimer (normal value 0.27–0.77 µg/mL) in plasma samples. Drug-related adverse events (AEs) leading or not to RBV discontinuation were also considered. 

All PTs underwent standard-of-care treatment according to local guidelines. 

This study was conducted in accordance with the ethical principles of the current Declaration of Helsinki and in accordance with International Conference on Harmonisation Good Clinical Practice (ICH GCP), Good Epidemiology Practices (GEPs), and applicable regulatory requirements. Samples were collected with informed consent in accordance with the Helsinki Declaration and with local ethical committee approvals: Covid-BioB, ClinicalTrials.gov NCT04318366; Ethical Committee approval number: 34/int/2020. 

### 2.4. Outcomes

The primary endpoint was to assess the effect of aerosol RBV on SARS-CoV-2 viral load in nasopharyngeal samples, expressed as the proportion of participants achieving a “virus-negative” status by means of real-time polymerase chain reaction (PCR) at discharge. The secondary endpoints were to measure the dynamics of SARS-CoV-2 in nasopharyngeal samples by using a semi-quantitative real-time polymerase chain reaction (PCR) assay threshold cycle (Ct) at baseline (BL), at the end of treatment (EOT), and at discharge. In some PTs who were discharged with a SARS-CoV-2-positive PCR, a subsequent specimen was investigated within 14 days of discharge. Another secondary endpoint was the assessment of clinical efficacy as expressed by the change in the clinical status severity scale (CSS), calculated on a 7-point ordinal scale from the first dose date (baseline evaluation, BL) to 6 days (EOT) or to the date of RBV aerosol discontinuation and discharge or death.

### 2.5. Sample Collection and Virological Assays

Sequential nasopharyngeal samples (taken at least on day 1 corresponding to BL evaluation and day 6, EOT) were aliquoted and stored at −80 °C until testing. Identification and semi-quantitative analysis of SARS-CoV-2 were performed through the Cobas^®^ SARS-CoV-2Test (Roche Diagnostics) using the fully automated Cobas^®^ 6800/8800 Systems under FDA Emergency Use Authorization (EUA). The assay is a single-well dual-target assay, which includes specific detection of SARS-CoV-2 by targeting conserved regions within the ORF 1a/b and E genes, a full-process negative control, positive control, and internal control. Results were recorded both as positive/negative and semi-quantitatively based on amplification cycle threshold (Ct) values for one of the two target genes. A decrease in viral load corresponded to an increase in Ct value; a Ct ≥ 39.00 was considered negative. 

The mutational profile was assessed by sequence analysis of SARS-CoV-2 whole-genome sequencing (WGS) of nasopharyngeal samples, as previously described [21]. Briefly, RNA extracts were processed with the CleanPlex^®^ SARS-CoV-2 Panel (Paragon Genomics, Hayward, CA, USA) and sequenced with MiSeq Reagent Kit v2 (300-cycles) (Illumina, San Diego, CA, USA) on the Illumina^®^ MiSeq platform. Genomic reconstruction was performed using the SOPHiA DDM^™^ platform (SOPHiA Genetics, Lausanne, Switzerland). The mutational spectrum was inferred on genome consensus sequences by using the Stanford Coronavirus Resistance Database (CoV-RDB; https://covdb.stanford.edu). A maximum-likelihood phylogenetic tree containing the SARS-CoV-2 sequences was constructed by using the Clustal Omega of EMBL site: Bioinformatics Tools for Multiple Sequence Alignment < EMBL-EBI. The nucleotide distances were calculated by generating a distance matrix using the maximum-likelihood model in the DNADIST program (PHYLIP 3.5c package). The phylogenetic tree was drawn using TreeViewPPC version 1.5.3.

### 2.6. Statistical Analysis

The Friedman one-way repeated-measure analysis of variance was performed with SAS Software (version 9.4) to analyse changes in Ct and CSS values over time. The Wilcoxon signed-rank test was used to assess significant changes at discharge from baseline in biochemistry and other parameters.

## 3. Results

### 3.1. Clinical Features in the Study Group

Changes in SARS-CoV-2 viral load over time and parameters associated with COVID-19 disease severity are summarised in Figure 1 and Table 1.

A total of nine PTs with mild-to-moderate COVID-19 pneumonia were included: eight males and one female. The median age was 53 years (IQR 51–64). The most common pre-existing comorbidities were cardiovascular/metabolic diseases: PT1, PT3, PT4, and PT7 suffered hypertension controlled by optimal antihypertensive therapy; PT2 and PT4 had dyslipidaemia. Participants 2 and 3 also had type 2 diabetes mellitus. Two PTs (PT8 and PT9) had hematologic malignancy (PT8 suffered from G2 follicular lymphoma, and PT9 from diffuse bulky large B-cell mediastinal lymphoma), while PT5 and PT6 had no comorbidities. At baseline, the median percentage of lung involvement determined by CT scans was 40% (IQR 30–45%). Seven PTs (PT1-PT5, PT8, and PT9) had a CSS value of 4. Two PTs (PT6, PT7) had a CSS value of 3. The median SpO_2_ value at BL under different oxygen flow rates was 94 (IQR 92–96), with a median PaO_2_/FiO_2_ ratio of 162 mmHg (IQR 143–228 mmHg). 

At EOT, four out of nine PTs (PT1-4) experienced an improvement in their CSS, three (PT6, PT7, PT9) had no change in CSS, and the remaining two (PT8 and PT5) had a worsening of their CSS scale points. The SpO_2_ ameliorated in five PTs, remained unchanged in two, and slightly worsened in two other PTs, while PaO_2_/FiO_2_ increased in seven out of nine PTs (Figure 1 and Table 1). 

At discharge, all these parameters reflecting clinical status improved in seven PTs with a favourable outcome, while they worsened in two PTs (PT8, PT9) who died during hospitalisation (Figure 1 and Table 1).

Changes in RNA Ct values (*p* = 0.032) and CSS values (*p* = 0.091) over time from baseline were assessed by the Friedman test; changes from baseline in lung involvement (*p* = 0.875), SpO_2_ (*p* = 0.405), and PaO_2_/FiO_2_ (*p* = 0.110) were analysed by the Wilcoxon signed-rank test. The middle part of each box represents the median (second quartile). The upper and lower limits of each box represent the first quartile (Q1) and the third quartile (Q3), respectively. The lines extending from each box (whiskers) represent the range of the data. 

Among the seven PTs with a favourable outcome, a decrease in lung involvement was observed in four, (PT1-4), while lung damage remained unchanged in two PTs (PT6, PT7) and slightly increased in the remaining one (PT5, Table 1). In the two PTs (PT8 and PT9) who died, a progressive worsening of clinical status was observed because of superinfection (PT8 had *Staphylococcus aureus* and *Aspergillus fumigatus* pulmonary superinfection) or progression of the underlying haematological malignancy (PT9) (Table 1). 

### 3.2. Biochemistry in the Study Group

Biochemistry at BL and discharge are described in Table 2. At BL evaluation, most participants had lymphopenia with a median total lymphocyte count of 0.9 × 10^9^/L (IQR 0.6–1.4) and a median ferritin level of 2266 ng/mL (IQR 1038–3391). Other markers of inflammation or cytokine activation like C-reactive protein, fibrinogen, lactate dehydrogenase, and interleukin-6 (IL-6) values were found abnormally elevated (Table 2). At discharge, total lymphocytes were significantly increased (*p* = 0.012), and biomarkers of inflammation (ferritin, C-reactive protein, fibrinogen, and lactate dehydrogenase) decreased, while the IL-6 value continued to increase with a median value of 15.9 picogram/mL (IQR 5.9–30.2) (Table 2). The median time from BL to discharge or death was 13 days (IQR 10–16). None of the PTs had adverse events (AEs) potentially associated with RBV therapy.

### 3.3. Concomitant Treatment

During hospitalisation, the nine PTs received low-molecular-weight heparin and, with the exception of PT8, intravenous glucocorticoid treatment. Two PTs (PT1 and PT2) received, in accordance with local protocols/guidelines, the anti-interleukin-1 agent, anakinra, at a dose of 10 mg/Kg/die for a total duration of 10 days; three PTs (PT6, PT7, and PT8) received the anti-IL-6 agent, tocilizumab, at a dose of 8 mg/Kg according to local therapy protocols and AIFA (Italian drug agency) approval. 

Targeted antibiotic therapy for the onset of bacterial superinfection was administered to three out of nine PTs (PT3, PT8, PT9). One participant (PT3) received remdesivir for 5 days and hyperimmune plasma infusion before RBV therapy. 

### 3.4. Virological Findings

Overall, the median Ct values progressively decreased during the period of observation, including the two PTs (PT8 and PT9) who died during hospitalisation: the median Ct value was 22.4 (IQR 19.84–5.07) at BL, 26 (IQR 23.34–34.06) at EOT and 27.92 (IQR 26.43–36.11) at discharge. Only one PT (PT2) had an undetectable viral load at EOT (≥39 with reappearance of viral RNA at discharge, but with a high Ct, indicating low viral replication). Differences in Ct values among the considered time points (BL, EOT, discharge) were found to be statistically significant; *p* = 0.032 (Figure 1). 

Considering the PTs with a favourable outcome, virus clearance was observed in two out of seven (28.6%, PT4 and PT7) at discharge, and the cumulative clearance rate was 71.4% within 14 days from discharge, because three other PTs had a negative RNA test at a follow-up visit after discharge.

A consensus sequence of SARS-CoV-2 by whole genome sequencing (WGS) was available in five out of nine PTs at BL and EOT. In three out of five PTs (PT3, PT8, PT9), WGS was also available in at least one other time point before BL evaluation. In total, 15 SARS-CoV-2 genomes were analysed. The sequences from sequential nasopharyngeal samples of these five PTs were analysed by inferring Stanford Coronavirus Resistance Database (CoV-RDB; https://covdb.stanford.edu, accessed on 21 September 2023) for the definition of VOCs and viral lineages.

In detail, nasopharyngeal samples from PT3 belonged to the Alpha variant, lineage B.1.1., samples from PT6 and PT7 clustered with the Delta variant, lineage AY75, while PT8 and PT9 were infected by the Omicron variant, lineages BA.2 and BA.1.17, respectively. The phylogenetic tree construction showed that each sequence clustered within the respective variant and was related to each other virus in the same PT (Figure 2).

In three of these five PTs, sequences were also available in at least one other time point before BL evaluation (T1 or T1 and T2). The phylogenetic tree was constructed with reference sequences identified as the best-matched sequence for each PT consensus sequence by using the Stanford Coronavirus Resistance Database (CoV-RDB; https://covdb.stanford.edu); accession number: EPI_ISL 2762232 (Alpha variant); EPI_ISL 2793575 and EPI_ISL 6855120 (Delta variant); OM371884.1 and OV310583.1 (Omicron variant); NC045512.2 (Wuhan isolate).

Samples from four out of five PTs (PT3, PT6, PT7, PT8) exhibited very closely related sequences at different time points investigated, while samples from the remaining PT (PT9) had an interesting profile: the EOT sequence was more closely related to those obtained at T1 and T2 (3 and 2 months before RBV administration, respectively) than to the sequence obtained at BL. Notably, this PT had a long-lasting SARS-CoV-2 infection with hospitalisation in April 2022.

Sequences of the five PTs were also analysed for their amino acid (aa) mutational profile along different time points. Data on aa change at different time points are summarised in Appendix A. All aa substitutions were considered in each PT’s sample with respect to the best-matched isolate that was used for comparison (Appendix A). 

Samples from PT3, who was infected by the Alpha variant, and those from PT7, harbouring the Delta variant, showed several aa changes distributed along the whole genome when compared with the respective reference sequence (Appendix A). However, the sequence pattern in these two PTs remained unchanged at EOT with RBV (Appendix A). 

An interesting mutational spectrum was revealed along different time points in PT6, PT8, and PT9 with respect to the sequences used for comparison. 

In the nasopharyngeal sample of PT6, infected by the Delta variant, the G446V substitution, which is associated with reduced susceptibility to several neutralising monoclonal antibodies (mAbs), was revealed at EOT.

In PT8, an aa substitution in the spike region (K182N) was detected only at BL evaluation, while at EOT, the substitution G96V emerged in the nucleoprotein. 

An aa change, E155K within the nsp8, and two mutations within the spike protein, V67A and L455S, were revealed in PT9 at EOT but not in the samples obtained at the other time points (T1, T2, BL). Furthermore, the substitution C464F within nsp12 (RNA-dependent RNA polymerase) was present at BL but not in the other time points analysed (T1, T2, EOT). In this PT, the ORF7a showed a higher complexity in its mutational profile with the disappearance of some aa mutations and the appearance of other aa substitutions during the period of observation (Appendix A). In detail, at BL, the aa substitution T39I was revealed, while the aa change T111I emerged at EOT. 

The mutational pattern of these five PTs was also compared with the prototype Wuhan-Hu-1 (accession number NC_045512.2). Samples from participant 3 showed a deletion at position 144 (D144) within the spike protein, associated with reduced susceptibility to several mAbs [22]. Participant 7, harbouring the Delta variant, invariably displayed L452R, already described for its association with reduced susceptibility to bamlanivimab and a 5.7-fold reduction in susceptibility to cilgavimab. Several aa changes commonly present in the Omicron variant and associated with various degrees of reduced susceptibility to neutralising mAbs were detected in the samples belonging to PT8 [G142D, S371F, D405N, K417N, N440K, E484A, Q493R] and PT9 (Δ142–144, R346K, S371L, K417N, N440K, G446S, E484A, Q493R) [23].

## 4. Discussion

We retrospectively evaluated virological and clinical outcomes in adults hospitalised with COVID-19 pneumonia with the primary objective of assessing the effects of RBV aerosol on SARS-CoV-2 viral load. At discharge, we found virus clearance in two out of seven (28.6%) PTs with a favourable clinical course. The other five PTs with a good clinical outcome had very low viral replication at discharge, and three out of three PTs with an available follow-up sample achieved a virus-negative status within 14 days from discharge. Therefore, we had a cumulative viral clearance rate of 71% at the last follow-up visit. Additionally, several PTs, including the two PTs (PT8 and PT9) who deceased during hospitalisation, showed a significant decline (*p* = 0.032) in SARS-CoV-2 viral load in nasopharyngeal samples measured from BL to discharge (Figure 1). 

Published results on the use of ribavirin in the treatment of SARS are controversial [24]. 

Several studies have shown that RBV inhibits the replication of SARS-CoV in vitro [25,26,27]. However, other studies have found that RBV does not exert any effect on virus replication in vivo [28,29]. 

One study in patients hospitalised for severe COVID-19 showed that intravenous administration of RBV was not associated with improved negative conversion time for SARS-CoV-2 tests and improved mortality rate when comparing the experimental group with a control group who did not receive RBV. However, RBV was administered systemically, possibly resulting in a lower concentration of the drug at the site of infection with respect to administration via aerosol. Additionally, the study included patients with severe COVID-19, and the authors concluded that the role of RBV in patients with mild SARS-CoV-2 infection remains to be elucidated [30].

One clinical trial investigating the effect of IFN beta-1b, lopinavir–ritonavir, and ribavirin on negative conversion time in patients with mild-to-moderate COVID-19 showed that the combination group had a significantly shorter median time from the start of study treatment to negative nasopharyngeal swab (7 days) than the control group under lopinavir–ritonavir (12 days). They concluded that early triple antiviral therapy was safe and superior to lopinavir–ritonavir alone in shortening the duration of viral shedding [31].

A recent trial on RBV aerosol administration in hospitalised adults with respiratory distress for COVID-19 showed negative SARS-CoV-2 tests in 24/28 (85.7%) patients who completed the trial (30-day assessment) [32]. Another study from China on critically ill COVID-19 patients who received RBV orally showed virus clearance in 63% of patients on day 21 of treatment with a significant decrease in viral load when comparing day 21 to the baseline for RBV therapy [33].

Our previous report within a compassionate use study of RBV aerosols in patients with SARS-CoV-2 infection showed a negative SARS-CoV-2 test in five out of five participants at the end of the quarantine period (day 14 after hospital discharge) [10]. Altogether, these previous studies and the present report suggest that RBV aerosols limit SARS-CoV-2 replication, which may play a role in the prevention and/or attenuation of damage exerted by local and systemic inflammation. Of note, 89% of PTs in our study were on treatment with corticosteroids, which might reduce the antiviral effect of RBV but also the inflammatory response [34,35]. 

Concerning the clinical status, an improvement in the CSS scale showing a trend toward significance (*p* = 0.091) was observed in the study group. Respiratory function parameters (SpO_2_, PaO_2_/FiO_2_) were found to be ameliorated at the end of treatment, although they did not reach statistical significance, with a decrease in lung involvement also measured at discharge. The markers of inflammation, in particular fibrinogen, showed a declining trend at discharge (Table 2) as well as a significant increase in total lymphocytes (*p* = 0.012), the latter parameter reflecting disease severity [36].

The study by Poulakou et al., investigating the potential of RBV inhalation solution to reduce COVID-19 disease severity in adults with a diagnosis of respiratory distress, showed an improvement of clinical severity status rating in 31.4% (16/51) of patients at the end of treatment and 78.4% (40/51) of patients at a day 30 assessment [32]. 

One other study from China, including severe and critical COVID-19 patients who received RBV orally, showed 100% survival [33]. Since the viral load of SARS-CoV-2 in the lower respiratory tract declined with the use of RBV, the authors suggested that the decrease in viral load positively influenced the survival rate. 

Eslami G. et al., comparing oral RBV vs. sofosbuvir/daclatasvir combination treatment, showed a better outcome in the sofosbuvir/daclatasvir arm vs. the RBV arm [37]. However, this study used a combination of two potential antivirals vs. RBV alone, with the latter compound administered at a high dosage. 

Notably, high doses of oral RBV have many adverse effects like anaemia and may impair renal function, which might complicate advanced cases of COVID-19. In this regard, the authors suggested that the relative advantage of the combination treatment with sofosbuvir/daclatasvir could be the lack of excess adverse events compared to the RBV arm. 

Regarding optimal dosage via aerosol, the dose of ribavirin used in this study, 1000 mg per treatment (administered as 10 mL of 100 mg/mL), was based on the findings from preclinical studies and clinical trials and is consistent with the dosing in the COVID-19 compassionate use study [10]. A randomised, placebo-controlled study evaluated the safety and pharmacokinetics of inhaled ribavirin [38]. Doses ranged from 50 to 100 mg/mL, delivered in a single inhalation of either 20 or 40 min duration. Ribavirin absorption reached *C*_max_ within 2 h across cohorts. Four single-dose regimens of ribavirin aerosols demonstrated systemic exposure with minimal systemic effects, indicating a favourable dose–exposure–effect relationship for ribavirin aerosols that may encourage their use in the coronavirus clinical setting.

The delivery regimen of ribavirin by aerosol was well tolerated in our participants with no systemic AEs such as anaemia or renal impairment. However, blood samples for pharmacokinetic analysis were not collected in the current study. So, the systemic exposure to ribavirin was unknown. 

We then investigated the mutational profile of SARS-CoV-2 in five PTs with available specimens, before and after the 6-day course of RBV (EOT), because in vitro studies indicated an inhibitory and mutagenic effect of RBV on SARS-CoV-2 [25,39]. Three of these five PTs had a favourable outcome and two (both PTs with concomitant hematologic malignancy) had a fatal outcome. 

Of note, these five PTs were infected during different waves and therefore harboured different variants: one PT (PT3) was infected by the Alpha variant (lineage B.1.1.7); two other PTs (PT6 and PT7) by the Delta variant (lineage AY75), and the remaining two PTs (PT8 and PT9) by the most recent Omicron variants (lineage BA.2 and BA.1.17, respectively; Appendix A). 

The phylogenetic analysis of whole genomes showed that sequential nasopharyngeal samples from each PT, except PT9, were closely related although not identical (Figure 2). Interestingly, in PT9 with a long-lasting SARS-CoV-2 infection, the EOT sequence was more closely related to that obtained at T1 and T2 (3 and 2 months before RBV administration, respectively) than to the sequence obtained at BL (Figure 2). However, considering the phylogenetic tree of all samples investigated, RBV seemed to have no or little mutagenic effect on the whole SARS-CoV-2 genome. 

Regarding the mutational profile of SARS-CoV-2 during RBV treatment, in two PTs (PT3 and PT7, harbouring the Alpha and Delta variants, respectively), the variant detected in the nasopharyngeal sample before RBV administration remained unchanged in the subsequent samples. This finding indicates that in these two cases, even in the presence of RBV, no novel mutations were fixed during 6 days of RBV treatment. 

Notably, PT3 received a 5-day course of remdesivir (RDV) before RBV. However, sequence analysis of nasopharyngeal samples did not show a clear effect of RDV on viral evolution by comparison of mutational patterns at T1 (before remdesivir administration), T2 (corresponding to first administration of remdesivir), and BL administration of RBV (corresponding to the completion of a 5-day course of RDV treatment) (Appendix A). In this regard, a recent study showed no resistance mutations and minimal intra-host diversity in serial samples of patients receiving short courses of RDV for moderate COVID-19 disease, suggesting that the barrier to RDV resistance is high in patients with moderate disease under short-term treatment [40]. 

Interestingly, PT6, PT8, and PT9 showed aa mutations at EOT with respect to BL evaluation. These mutations were not located in the nsp14 exonuclease domain, which is an active part of the replication complex and increases the fidelity of the nsp12 viral RNA polymerase (RdRp), nor within the nsp12, where an effect of RBV is expected [41,42]. 

Whole-genome sequencing of PT6 revealed an aa change (G446V) within the spike protein at EOT. Sequences from PT8 showed an aa change at EOT within the nucleoprotein (G96V), while sequences from PT9 showed an alternation of single mutations. At EOT, an E155K aa substitution within the nsp8, two mutations within the spike protein, V67A and L455S, and two aa mutations (T111I and I110T) in the ORF7a appeared that were not present at BL. One of these mutations (I110T) was detected about 2 months before RBV but disappeared at BL of RBV treatment, while the other mutation, T111I, was revealed after a short period (day 6 of RBV administration). This PT had a prolonged infection that could have generated new SARS-CoV-2 variants. 

However, most of the mutations in these three PTs were detected after a short period from the first RBV administration (EOT, corresponding to the completion of a 6-day course of RBV treatment). Since the mutation rate of SARS-CoV-2 is high (10^4^ to 10^6^ substitutions per nucleotide per round of replication) in response to selective pressure, we cannot exclude that the intra-host emergence of SARS-CoV-2 variants was consequent to pressure, although minimal, exerted by RBV [43,44].

Individual mutations within the spike protein that reduce monoclonal antibody neutralisation efficacy were invariably detected in four out of five PTs with available sequential nasopharyngeal samples. In detail, the deletion at position 144 (Δ144) that is associated with resistance to several NTD-binding neutralising antibodies [22] was revealed in sequential samples from PT3 infected by Alpha variant B.1.1.7. Participant 7, harbouring the Delta variant, displayed L452R, associated with reduced susceptibility to bamlanivimab as well as cilgavimab. Several aa changes commonly present in the Omicron variants and associated with various degrees of reduced susceptibility to neutralising mAbs were detected in serial samples of PT8. Whole-genome sequencing of sequential samples obtained from PT9 showed the aa substitution G446S, which induces resistance to tixagevimab and cilgavimab [45]. Notably, he received treatment with these two mAbs about two months before RBV, with no benefit.

However, the mutational spectrum associated with resistance to several mAbs was unchanged after RBV treatment.

Determining the extent to which variants reduced mAbs susceptibility is critical to preventing and treating COVID-19. So, our data raise the importance of accumulating new data on the genetic characterisation of SARS-CoV-2 variants to better understand the place of antiviral treatment in people with mild-to-moderate COVID-19 that may be less or not responsive to mAbs. 

The main limitation of this study is its retrospective nature including a small group of PTs with similar baseline pulmonary features but in the context of a heterogeneous clinical background, which in two cases influenced an unfavourable clinical outcome.

One other important limitation is that we could not firmly assess the role of RBV on clinical improvement nor on virological outcome because we did not include a placebo control group. Additionally, treatment of COVID-19 in hospitalised patients included multiple medical therapies. Because additional medications (e.g., corticosteroids, immunomodulatory molecules, and other antivirals) were permitted in this study of ribavirin aerosols, it is not possible to identify the relative contribution of each agent to the overall therapeutic effect. 

Finally, our PTs were infected by different SARS-CoV-2 variants. Therefore, clinical and virological outcomes could be influenced by differences in the severity of disease in addition to the BL mutational pattern [46,47].

In conclusion, the decline and the clearance of SARS-CoV-2 RNA occurred in a relatively short period after RBV administration in our group of PTs with mild–moderate COVID-19 pneumonia, suggesting that RBV, in addition to a supportive treatment approach, favoured a better clinical outcome and contributed to the mutational spectrum of SARS-CoV-2 in some cases. Our results encourage extending clinical trials to understand the place of RBV, especially in combination therapies in view of new emerging SARS-CoV-2 variants that could severely affect current vaccination strategies and therapeutics. 

## Figures and Tables

**Figure 1 microorganisms-12-01146-f001:**
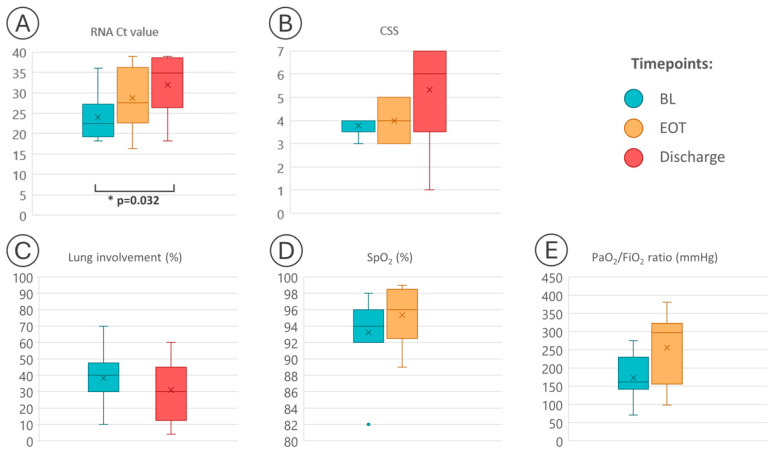
Boxplot summary of SARS-CoV-2 RNA dynamics (**A**), change in clinical status severity (**B**), and other clinical parameters ((**C**), lung involvement; (**D**), SpO_2_; (**E**), PaO_2_/FiO_2_) in participants who received ribavirin aerosol for COVID-19 pneumonia. Abbreviations: BL, baseline (day 1, immediately before ribavirin administration); EOT, end of treatment with ribavirin (day 6); Ct, cycle threshold; CSS, clinical status severity; SpO_2_, oxygen saturation; PaO_2_, arterial oxygen partial pressure; FiO_2_, fractional inspired oxygen. A Ct value ≥ 39.00 is considered a negative result.

**Figure 2 microorganisms-12-01146-f002:**
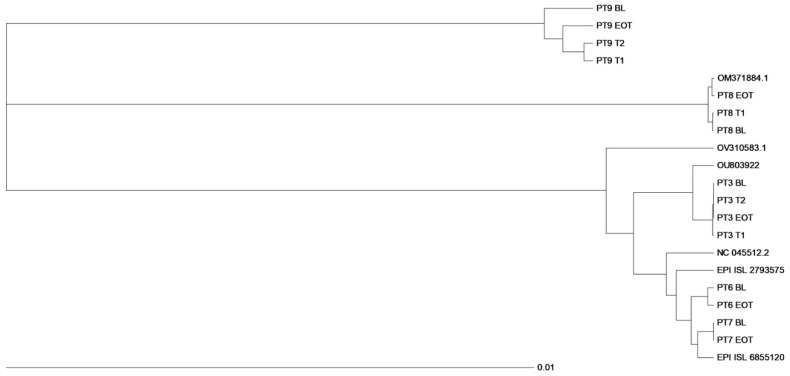
Phylogenetic tree of SARS-CoV-2 whole genome according to different time points (baseline, BL, and end of treatment, EOT) in 5 participants who received RBV aerosol therapy. A maximum-likelihood phylogenetic tree containing the SARS-CoV-2 isolates obtained from nasopharyngeal samples was constructed using Clustal_Omega on the EMBL site: Bioinformatics Tools for Multiple Sequence Alignment < EMBL-EBI.

**Table 1 microorganisms-12-01146-t001:** Clinical and virological findings at baseline, end of treatment, and discharge in participants who received ribavirin aerosol for COVID-19 pneumonia.

PT	Age (yrs)	Sex	BL					EOT				Discharge			Death
			RNA Ct	Lung Involvement (%)	CSS	SpO_2_ (%)	PaO_2_/FiO_2_ Ratio (mmHg)	RNA Ct Value	CSS	SpO_2_ (%)	PaO_2_/FiO_2_ Ratio (mmHg)	RNA Ct Value	Lung Involvement (%)	CSS	
**1**	66	M	22.40	40	4	96	275	34.85	5	98	335	38.00	10	7	No
**2**	73	M	29.29	10	4	94	145	39.00 ^1^	5	94	300	37.44	4	7	No
**3**	52	F	19.84	70	4	92	232	24.48	5	89	267	27.92	40	6	No
**4**	53	M	36.09	45	4	96	143	33.79	5	95	381	39.00 ^1^	15	7	No
**5**	51	M	24.94	30	4	93	163	21.21	3	96	102	34.78	40	6	No
**6**	34	M	25.07	30	3	98	162	37.45	3	99	210	27.52	30	6	No
**7**	64	M	18.59	30	3	92	71	27.52	3	99	311	39.00 ^1^	30	7	No
**8**	54	M	21.82	40	4	96	228	16.30	3	91	98	18.24	60	1	Yes
**9**	31	M	18.13	50	4	82	141	24.05	4	97	297	25.35	50	1	Yes
**Q2**	53	NA	22.40	40	4	94	162	26.00	4	96.5	297	27.92	30	6	NA
**Q1**	51	NA	19.84	30	4	92	143	23.34	3	94.75	210	26.43	15	6	NA
**Q3**	64	NA	25.07	45	4	96	228	34.06	5	98.25	311	36.11	40	7	NA
***p*-value**								-	-	0.405 ^2^	0.110 ^2^	0.032 ^3^	0.875 ^2^	0.091 ^3^	

^1^ A cycle threshold value ≥ 39.00 is considered a negative result. ^2^ By Wilcoxon signed-rank test. ^3^ By Friedman analysis. Abbreviations: PT, participant; BL, baseline (day 1, immediately before ribavirin administration); EOT, end of treatment with ribavirin (day 6); Ct, cycle threshold; CSS, clinical status severity; SpO_2_, oxygen saturation; PaO_2_, arterial oxygen partial pressure; FiO_2_, fractional inspired oxygen; Q2, median value; Q1, first quartile; Q3, third quartile; NA, not applicable.

**Table 2 microorganisms-12-01146-t002:** Biochemistry at baseline and discharge in participants who received ribavirin aerosol for COVID-19 pneumonia.

PT	Baseline Date	WBCs (×10^9^/L)	Total Lymphocytes (×10^9^/L)	Ferritin (ng/mL)	IL-6 (pg/mL)	Lactate Dehydrogenase (U/L)	C-Reactive Protein (mg/L)	Fibrinogen (ng/dL)	D-Dimer (µg/mL)	Date of Discharge	WBC (×10^9^/L)	Total Lymphocytes ×10^9^/L)	Ferritin (ng/mL)	IL-6 (pg/mL)	Lactate Dehydrogenase (U/L)	C-Reactive Protein (mg/L)	Fibrinogen (ng/dL)	D-Dimer (µg/mL)
**1**	24/03/2021	7.5	0.7	2266	42.7	544	84.2	723	0.75	31/03/2021	4.9	1.5	1484	8	250	1.1	477	0.96
**2**	15/04/2021	8.1	1.4	1719	9.7	386	51.4	597	2.61	23/04/2021	8.6	2.4	700	4.2	207	1.4	360	0.52
**3**	04/06/2021	6.4	2.3	497	20.4	176	17.4	634	1.11	17/06/2021	5.6	2.4	200	5.9	216	2.9	481	0.63
**4**	10/09/2021	7	1.7	3259	14.7	354	19.1	491	0.56	24/09/2021	6.7	1.8	806	2.7	215	1.3	437	0.27
**5**	27/10/2021	2	0.6	11,685	10	1187	12.7	529	0.66	11/11/2021	6.4	1.9	1272	347	267	0.2	333	0.27
**6**	20/11/2021	5.4	1	1038	13.1	391	30.2	645	0.58	03/12/2021	9.5	3.3	637	15.9	239	0.6	271	0.27
**7**	24/11/2021	6.6	0.9	814	2.4	525	77.5	644	0.78	04/12/2021	7.1	2.2	471	108	307	0.6	224	0.27
**8**	12/05/2022	9.9	0.1	3391	15.9	405	8.7	653	0.3	02/06/2022	10.8	0.1	32,388	15.9	900	146.1	736	4.41
**9**	11/08/2022	3.3	0.2	6448	10.8	831	30.4	313	1.83	27/08/2022	3.2	0.2	5834	30.2	598	332.3	532	2.82
**Q2**		6.6	0.9	2266	13.1	405	30.2	634	0.75		6.7	1.9	806	15.9	250	1.3	437	0.52
**Q1**		5.4	0.6	1038	10	386	17.4	529	0.58		5.6	1.5	637	5.9	216	0.6	333	0.27
**Q3**		7.5	1.4	3391	15.9	544	51.4	645	1.11		8.6	2.4	1484	30.2	307	2.9	481	0.96
***p*-value (baseline vs. discharge) ^1^**											0.173	0.012	0.110	0.906	0.109	0.515	0.066	0.515

^1^ By Wilcoxon signed-rank test. Abbreviations: PT, participant; BL, baseline; WBCs, white blood cells; Q2, median value; Q1, first quartile; Q3, third quartile.

## Data Availability

The data presented in this study are available on request from the corresponding author.

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
