# Peer review of "Clinico-Virological Outcomes and Mutational Profile of SARS-CoV-2 in Adults Treated with Ribavirin Aerosol for COVID-19 Pneumonia"

_microorganisms, 2024, doi:10.3390/microorganisms12061146_

Round 1

Reviewer 1 Report

Comments and Suggestions for Authors

The manuscript refers to a pilot study of aerosolised ribavirin in 9 SARS coV-2 infected patients.  The study does not have a proper clinical design since several important parameters have been left out, such as the uptake of the drug by the mucosa and its clearance. There is no clear data on the optimal dose based on proper studies. In addition, there are no controls that may be useful in determining the pharmacodynamics of the drug.  In addition, the subjects chosen for the study were heterogeneous in infection, SpO2 parameters, and several other issues. How do the authors certify that the effects of the drug on viral infection are as they described? By analysing limited viral replication? There is no follow-up after patient discharge, ie possible reinfection. The manuscript does not have the high-quality standards required to be published in this journal.

Comments on the Quality of English Language

There are several grammatical mistakes in the text.

Author Response

In response to Reviewer 1.

-We agree with the Reviewer that a pharmacodynamic study measuring the uptake of ribavirin by the mucosa and possibly its clearance, could be of great interest. 

However, this is a retrospective analysis from a compassionate study, where bronchoalveolar specimens were not obtained, before, during and after RBV administration to perform pharmacodynamic and pharmacokinetic of the drug in this compartment.  This type of analysis should be performed in a randomized clinical trial or in preclinical studies on animal models (see also the trial by Couroux P, doi:10.1111/cts.13350 and the study by Gilbert BE, doi:10.1016/j.antiviral.2008.01.005).  However, we provided references on several preclinical and clinical research evaluating the optimal aerosol dosage of RBV for viral infections of the upper and lower respiratory tracts (reference 10, 25, 32, 38, 39). So, studies on the optimal dosage of RBV via aerosol, supporting the dosage and administration described in the present study are currently available.   In detail, the dose of ribavirin aerosol used in this study—1000mg per treatment (administered as 10 mL of 100mg/mL, was based on the findings from preclinical research and is consistent with the dosing in the COVID-19 compassionate use study (10 mL of 100mg/mL; Messina E, et al. doi:10.1007/s40121-021-00493-9). Although the long duration of administration recommended by the US Food and Drug Administration (FDA) for respiratory syncytial virus (20mg/mL administered continuously for 12– 18h/day) was an obstacle to the use of ribavirin aerosol, preclinical studies have shown that increasing the concentration of the ribavirin solution in the nebulizer reservoir could substantially reduce treatment time ( Gilbert BE, et al. doi:10.1016/j.antiviral.2008.01.005; Wyde PR, doi: doi:10.1016/0166-3542(87)90029-5.  Ribavirin was effective for preventing mortality ina lethal influenza A infection model: mice were treated with a solution of 100mg/ml in the nebulizer reservoir at a flow rate of 10 L of air per minute, which produced aerosol droplets that contained 2.3 mg of ribavirin per liter with a mass median aerodynamic diameter of 1.8 μm (Gilbert BE).

The pharmacokinetics and safety of aerosolized ribavirin has been evaluated in a single-dose study (500mg, 1000mg, or 2000mg of ribavirin administered at varying solution concentrations and treatment durations) in healthy volunteers. The pharmacokinetic profile of plasma ribavirin was linear across the dose range; for ribavirin doses of 10 ml of 50mg/ml (500mg), 20ml of 50mg/ ml (1000mg), 10 ml of 100mg/ml (1000mg), and 20ml of 100mg/ml (2000mg), area under the curve to the last measurable concentration of total ribavirin in plasma was 1.25, 2.54, 2.15, and 5.96μg/ml h, respectively, (Couroux P, doi:10.1111/cts.13350).

As suggested by the Reviewer, we added a comment in regard of optimal dosage of RBV, in the section Discussion, line 378-381 and related references (references 10 and 38).

-The decrease of viral load was statistically significant by comparing different time points: see, section Results, Figure 1. We analysed also other parameters of severity of the disease that ameliorated in all the participants after a short course of RBV administration via aerosol, although not all reached statistical significance (see Figure 1 and Table 1 in the new version of the manuscript).  The administration of antivirals, if received early, may have a positive impact on the clinical outcome, probably reducing not only viral load but also the inflammation that is consequent to viral spread into the respiratory tract. We also found an amelioration of respiratory and inflammatory parameters after a short course of RBV treatment. This finding is discussed in the section Discussion, lines 330-356.However, In the section, Discussion we included the limitations for this study (see line 448-459).

-After discharge the participants were followed for at least 14 days, and this data is available in the section, Material and Methods, sub-section, Outcomes, line 132,133, and section, Results, line 246,247 showing 78% virus clearance.

It is unclear, the request of monitoring these participants for reinfection, or maybe we have misunderstood the comment. However, regarding the possibility of SARS-CoV-2 reinfection, resilience and vaccination are the best approaches to prevent SARS-CoV-2 new infection.

To reduce the possibility of transmission of new incoming variants, by using antivirals in mild-moderate cases, may be a good strategy in regions with high endemicity.   This issue is discussed in the section, Discussion, line 443-446, with particular attention to the emergence of variants resistant to mAb.

Finally, we checked and tried to correct grammatical mistake in the text.

We hope to have sufficiently addressed and provided responses to the comments for this our study, as indicated by the Reviewer 1. 

Reviewer 2 Report

Comments and Suggestions for Authors

The manuscript ID microorganisms-2981307 titled "Clinico-virological outcomes and mutational profile of SARS-CoV-2 in adults treated with ribavirin aerosol for COVID-19 pneumonia" presents a crucial study given the current scenario. With the emergence of new SARS-CoV-2 variants potentially affecting vaccine efficacy, laboratory diagnosis, and available therapies, there is a growing interest in finding effective antiviral agents for SARS-CoV-2 infections. Ribavirin (RBV) is a broad-spectrum antiviral agent with demonstrated in vitro activity against various viruses, including SARS-CoV-2. This retrospective study evaluated the dynamics of viral clearance of SARS-CoV-2 in hospitalized adult participants (PTs) with COVID-19 pneumonia who received RBV aerosol within a compassionate-use study. The study also assessed the impact of RBV on clinical outcomes and the mutational profile of SARS-CoV-2. The median RNA values measured in nine PTs included in the study decreased from baseline to discharge (at BL, threshold cycle (Ct)=22.4, IQR 19.84-5.07; at discharge, Ct=27.92, IQR 26.43-36.11) with a significant decline of Ct value evaluated by Friedman rank ANOVA analysis, P=0.032. Seven out of nine PTs experienced clinical improvement, while two PTs deceased during hospitalization. In PTs with a favorable outcome, the virus clearance rate at discharge was 28.6%. The cumulative clearance rate was 71.4% within the 14th day of discharge. A mutational pattern after RBV treatment was detected in 3 out of 5 PTs for whom whole-genome sequencing data were available. The authors suggest that RBV limits SARS-COV-2 replication, possibly resulting in a favorable clinical outcome. However, ribavirin may also contribute to the mutational spectrum of SARS-CoV-2.

I commend the authors' efforts in conducting this study. However, several points need clarification before submitting a revision:

1. The statement regarding adverse events reported during RBV treatment should specify whether it pertains only to patients with COVID-19 or to those with other viral infections. Additionally, clarity on the age groups of the patients would be beneficial, considering that RBV is primarily used for infants and young children.

2. The mention of RBV's ability to limit viral replication based on multimodal antiviral properties should specify whether these properties are hypothetical or reported in the literature.

3. The statement regarding RBV causing transition mutations and potentially increasing the number of mutations in the viral genome raises concerns about the emergence of new SARS-CoV-2 variants or resistance to vaccines or drugs. Clarification on how this benefits patients would be helpful.

4. Define the parameter used to classify "primary mild-moderate COVID-19 pneumonia."

5. Clarify the dosing regimen of RBV aerosol, particularly the number of doses per day.

6. The methodological section appears convoluted; restructuring it with clear subsections could enhance understanding and aid in evaluating the results section.

Overall, the current version of the manuscript requires critical revision to address these concerns effectively.

Comments on the Quality of English Language

Minor editing of English language required

Author Response

In response to Reviewer 2

  1. We specified in the text that the AE were investigated specifically in adult participants with COVID-19 disease. However, as shown in studies on adults with other viral infections, RBV may have comparable AE (see, haemolytic anaemia after oral administration of RBV in hepatitis C chronic infection or administration via aerosol for respiratory syncytial virus). Several symptoms are of respiratory nature also during COVID-19 disease, and not related to drug administration. These symptoms are also common in Influenza or respiratory syncytial virus infection, where RBV was used.  In this regard, the administration of RBV via aerosol may have less frequency of AE because of low systemic distribution of this small molecule when administered via aerosol (see section , Discussion, line 371-374 and line 376-386). Concerning the age group, in the section, Introduction was detailed that RBV via aerosol is approved by FDA for respiratory syncytial virus in infant and children. However, it has been used also in adults with severe influenza virus infection (see section, Introduction, line 43-45, and references 7, 8) and for treatment of adults with SARS-CoV-2 infection (references 10, 32).  Additionally, all data on the procedures are available at gov NCT04318366, and the study was approved by local ethical committee Covid-BioB; Ethical Committee approval number 34/int/2020 where was specified the use of RBV in adults.  
  2. Antiviral activity of RBV are detailed in the section, Introduction with comments to previous studies, line 53-78 and related references, while in the section Discussion, line 328-254 are discussed in vitro and clinical studies regarding RBV activity on and SARS-CoV-2.
  3. Transition mutation that is one of mechanism for antiviral RBV activity, may be present along the full genome; however, we did not analyse the presence and frequency of transition in our sequences. We only add this mechanistic approach as a possible effect of RBV in the section, Introduction, line 60-65. Notably, we found several aa mutations along the spike protein that were present before the administration of RBV and could impair the response to mAb. This finding was discussed in the section, Discussion, line 443-446. Concerning the polymerase gene, we did not find any mutation emerging during RBV administration. This finding suggesting no activity of RBV on this domain, at least in our study population. However, we analyse specimens from a small number of participants.
  4. We added the parameters used to classify “primary mild-moderate” COVID-19 pneumonia in the section Materials and Methods, sub-section Study design and participants, line 83-85.
  5. We better explained the dosing regimen including the number of daily doses and the type of flow for aerosolization together with the equipment used (see section, Materials and Methods, sub-section, Procedures, line 93-99.
  6. The methodological section, Material and Methods was re-structured with sub-sections including: 2.1 Study design and participants; 2.2 Procedures; 2.3 Clinical and laboratory monitoring; 2.4 Outcomes; 2.5 Sample collection and virological assays; 2.7 Statistical analysis.

We hope to have addressed the concerns for this manuscript as suggested by the Reviewer 2.

Reviewer 3 Report

Comments and Suggestions for Authors

The material and methods section should be separated into sections or a figure of the experimental methodology.

In the case of PCR, was it homemade? Kit? Explain

Graph the data from Table 1 (means, percentages, standard deviation, do a comparative ANOVA) between BL, EOT, and Discharge.

The section on mutations in viral proteins should be represented with a figure. Showing mutations at different stages of treatment

Table S1 should be in the main text, where a comparative statistical analysis should be conducted and discussed between BL, EOT, and Discharge. 

Some of these laboratory data have been described as biomarkers of severity. They should be compared with the administration of ribavirin.

Author Response

In Response to Reviewer 3.

-The Materials and Methods were separated into sub-section. We hope to have ameliorated this section rendering the text easier to understand for the readers.  

-In the case of PCR, the assay we used to measure viral load was detailed in the section, Materials and Methods, subsection 2.5 Sample collection and virological assays as following: Identification and semiquantitative analysis of SARS-CoV-2 were performed through the Cobas® SARS-CoV-2Test (Roche Diagnostics) using the fully automated Cobas® 6800/8800 Systems under FDA Emergency Use Authorization (EUA). The assay is a single-well dual target assay, which includes specific detection of SARS-CoV-2 by targeting conserved regions within the ORF 1a/b and E genes, a full-process negative control, positive control, and internal control. Results were recorded both as positive/negative and semi-quantitatively based on amplification cycle threshold (Ct) values for one of the two target genes. A decrease in viral load corresponds to an increase in Ct value; a Ct>39 was considered as negative. (see line 138-147).

-The data in Table 1 were changed in a graphical representation and is now Figure 1 and statistical analysis was added according to parameters described in Figure 1.

-The section concerning the pattern of aa mutations along the whole SARS-CoV-2 genome was changed according to the suggestion by the Reviewer. So, Table S2 is now Figure S1.

-The Table S1 was moved in the main text adding a statistical analysis for all parameters investigated.

-We added statistical analysis as described above for parameters of disease severity including respiratory and inflammatory parameters. Some of these parameters were measured at baseline, end of treatment (EOT) and discharge, other were measured at baseline and EOT or discharge. These findings are described in Figure 1 and Table 1, with addition of statistical analysis.

We hope to have sufficiently addressed and provided responses to the comments for this our study, as indicated by Reviewer 3.  

Round 2

Reviewer 1 Report

Comments and Suggestions for Authors

The authors made important changes to the manuscript improving its quality. 

Author Response

In response to Reviewer 1

We thank the Reviewer for the time spent in reviewing this our study. We fully agree that the comments have greatly improved the quality of the manuscript.

We hope that the manuscript is now suitable for publication in the Journal Microorganisms.

Reviewer 3 Report

Comments and Suggestions for Authors

Line 382 remove the double "in"

Arrange the presentation of the tables and adapt them to the journal's format. Delete column "-" from table 1

Data from the statistical analysis carried out is needed.

Since Table 1 is repeated, check the table numbers and arrange them according to the text citation.

Comments on the Quality of English Language

Read the article again, as there are several spelling errors.

Author Response

In response to Reviewer 3.

We thank the Reviewer for the patience and accuracy in this second-round comments that have certainly improved the quality of the manuscript.

In detail:

-We apologize for the mistake, we deleted the “in” that was repeated in the text

- We modified the Tables (that are now Table 1 and Table 2) according with the journal’s format.

-Data from the statistical analysis was included in a Table, that is now Table 1, rather than in the text. This Table is in line and implements data of Figure 1. We chose to add a Table (Table 1), instead to include data in the text, because, on our opinion, it is now easier to understand the overtime change of different parameters as well as the clinical outcome after RBV administration.

So, Table 1 describing biochemical parameters is now Table 2. Normal values for laboratory parameters in footnote of Table 2 were deleted and added in the section Materials and Methods, subsection, Clinical and laboratory monitoring, line 113-117. So, Table 2 is now easier to read.

In Figure 1 was added an explanation concerning statistical analysis (see line 197-199).

-We checked the table numbers and tried to arrange them according to the text citation.

-We carefully read the manuscript and tried to correct spelling errors.

We made every effort to improve the quality of the manuscript according to the Reviewer’s comments.

We hope that the manuscript is now suitable for publication in the Journal Microorganisms.